# Nanocomposites of Chitosan/Graphene Oxide/Titanium Dioxide Nanoparticles/Blackberry Waste Extract as Potential Bone Substitutes

**DOI:** 10.3390/polym13223877

**Published:** 2021-11-10

**Authors:** Carlos Humberto Valencia-Llano, Moisés A. Solano, Carlos David Grande-Tovar

**Affiliations:** 1Grupo Biomateriales Dentales, Escuela de Odontología, Universidad del Valle, Calle 4B # 36-00, Cali 76001, Colombia; carlos.humberto.valencia@correounivalle.edu.co; 2Grupo de Investigación de Fotoquímica y Fotobiología, Facultad de Ciencias, Programa de Química, Universidad del Atlántico, Carrera 30 Número 8-49, Puerto Colombia 081008, Colombia; madolfosolano@mail.uniatlantico.edu.co

**Keywords:** bone tissue regeneration, chitosan nanocomposites, critical size defect, graphene oxide, titanium dioxide nanoparticles

## Abstract

New technologies based on nanocomposites of biopolymers and nanoparticles inspired by the nature of bone structure have accelerated their application in regenerative medicine, thanks to the introduction of reinforcing properties. Our research incorporated chitosan (CS) covalently crosslinked with glutaraldehyde (GLA) beads with graphene oxide (GO) nanosheets, titanium dioxide nanoparticles (TiO_2_), and blackberry processing waste extract (BBE) and evaluated them as partial bone substitutes. Skullbone defects in biomodels filled with the scaffolds showed evidence through light microscopy, scanning electron microscopy, histological studies, soft tissue development with hair recovery, and absence of necrotic areas or aggressive infectious response of the immune system after 90 days of implantation. More interestingly, newly formed bone was evidenced by elemental analysis and Masson trichromacy analysis, which demonstrated a possible osteoinductive effect from the beads using the critical size defect experimental design in the biomodels. The results of this research are auspicious for the development of bone substitutes and evidence that the technologies for tissue regeneration, including chitosan nanocomposites, are beneficial for the adhesion and proliferation of bone cells.

## 1. Introduction

Bone connective tissue is crucial to the human body, protecting vital organs, housing the bone marrow and minerals such as calcium and phosphate, supporting the body, and allowing its locomotion [1]. The global collective lawsuit for bone grafting grows daily due to many accidents and increased human osteoporosis, neoplasm, and osteoarthritis. Autografts are usually a common surgical practice for bone defects, removing material from the patient’s iliac crest, femur, or tibia [2]. 

Safety due to the absence of an immune response, histocompatibility, and the stimulation of the growth of new bone cells and their development (osteogenesis), make this technique attractive. On the other side, allografts include using bone from a donor (usually a cadaver) to repair and implant the bone-critical size defect [3]. 

Both autographs and allografts present osteoconduction, since their three-dimensional structure and bone morphogenetic proteins, together with various growth factors, facilitate the incorporation of vascularity and other osteoprogenitor cells through the differentiation of mesenchymal stem cells, stimulating bone renewal [4]. However, all these benefits are clouded by the report of numerous cases of chronic pain, bleeding, infection, and even patient morbidity [5]. A decrease in availability in conjunction with the previously reported problems of the gold standard techniques for bone grafting has generated a growing demand for safer and more practical solutions with sufficient availability. 

On the other hand, the use of biomaterials to mimic the biological, mechanical, and structural properties of connective tissues has grown significantly in the last decade by leaps and bounds [6]. Thanks to their similarity to the bone, many of these materials showed that they could stimulate genes for cell differentiation, extracellular matrix growth, and osteogenesis [3]. The design of three-dimensional and porous materials, organized at the microscopic and even the molecular level, allowed many investigations to demonstrate that biomaterials, including nanoparticles, can stimulate the growth of new bone and other connective tissues [7,8,9].

Even though promising biomaterials such as polysaccharides and proteins have been used for bone regeneration [10], there is still a great need to develop materials that meet all the requirements to be considered scaffolds for bone regeneration due to their low mechanical resistance. Biocompatible synthetic polymers such as polylactic acid (PLA) [11], poly ε-caprolactone (PCL) [12], polyvinyl alcohol (PVA) [13], among others, have even been used for tissue regeneration, with promising results. However, the selection of synthetic polymers has concerns related to environmental contamination and low-rate biodegradability. For this reason, the use of biopolymers reinforced with nanomaterials and nanofillers has been introduced in the last few years [14].

Chitosan is a derivative obtained from the chemical modification of chitin, a polysaccharide abundant in nature (mycelium of fungi, exoskeleton of crustaceans and insects) as cellulose or starch [15]. Chitosan has biological properties that meet the criteria required to prepare scaffolds that stimulate bone cells and new tissue growth and differentiation [16]. Many investigations reported excellent results in tissue regeneration using chitosan as a material for the construction of scaffolds. However, chitosan reinforcement with nanomaterials and natural extracts such as the blackberry extract (BBE) would guarantee biocompatibility, excellent durability, and better mechanical resistance, which are essential requirements in the regeneration of bone tissue not fulfilled yet. In this sense, incorporating graphene oxide (GO) and titanium dioxide (TiO_2_) nanoparticles would provide a greater contact surface and more excellent mechanical resistance without affecting biocompatibility, as we previously demonstrated in subcutaneous tissue studies [17]. Motivated by the observed biocompatibility in the previously prepared chitosan beads [18], we decided to study the possible stimulating effect for bone tissue regeneration of the chitosan beads implanted in critical bone defects of Wistar rats during 90 days of implantation. The objective was to determine by histological and SEM-EDS studies whether the CS/GLA/GO/TiO_2_/BBE beads can promote bone neoformation under critical size defect experimental design, which to the best of our knowledge, has not been reported yet with chitosan/graphene oxide/titanium dioxide beads. The present work results demonstrated that CS/GLA/GO/TiO_2_/BBE beads are potential bone tissue substitutes and are an open a window for new nanocomposites, including nanoparticles and natural extracts with durability and stability under physiological conditions. 

## 2. Materials and Methods

### 2.1. Materials

Chitosan (CS) (deacetylation ≥ 75%; medium molecular weight, Merck KGaA, Darmstadt, Germany), graphite (325 mesh, Alfa-Aesar, Tewksbury, MA, USA) for GO synthesis following Marcano’s procedure [19], Titanium isopropoxide (TTIP; reagent grade, 99%, Aldrich, Palo Alto, CA, USA) for TiO_2_ nanoparticles synthesis as reported previously [20]. GLA for crosslinking CS was provided by (Fischer Chemical, Philadelphia, PA, USA). The blackberry residues were obtained from a fruit and vegetable pulper distributor located in the Santa Elena neighborhood of the city of Cali, in the southern region of Valle del Cauca, at coordinates 3°27′00″ N, 76°32′00″ W. The residues consisted of a mixture of seeds, peel, and pulp of the fruit, preserved in a −20 °C freezer (Barnstead/Lab-line, Waltham, MA, USA) until use. The extraction and characterization procedures were presented in a recent publication [21].

### 2.2. Synthesis of Chitosan Beads

CS/GLA; CS/GLA/GO; CS/GLA/TiO_2_; CS/GLA/GO/TiO_2,_ and CS/GLA/GO/TiO_2_/BBE beads (Table 1) were synthesized and characterized elsewhere [18]. The different formulations used were:

### 2.3. Critical Bone Defect Studies

For the study, 15 biomodels (male Wistar rats, three months old and 380 g of average weight) were used in an experimental design of critical size bone, with defects of 5 mm × 0.8 mm (diameter × deep) in parietal bones. According to the five formulations of beads used (Table 1), the biomodels were organized into five groups of three animals. Two intraosseous preparations were made for each biomodel; in one, the corresponding bead formulation was implanted, and in the other, it was left as an empty defect control. The preparations were made with a trephine bur with an external diameter of 5 mm at low speed and constant irrigation with physiological saline. Anesthetic medications used were an intramuscular solution of Ketamine 70 mg/kg (Holliday Scott S Laboratory, Buenos Aires, Argentina) and Xylazine 30 mg/kg (Laboratorios ERMA, Celta, Colombia).

After 90 days of implantation, the biomodels were euthanized by an intraperitoneal application of 0.3 mL of Eutanex™ (Euthanex-INVET, Medellín, Colombia) consisting of 390 mg Pentobarbital sodium and 50 mg of diphenylhydantoin sodium/mL of saline solution. Subsequently, the samples were recovered and fixed in buffered formalin for 48 h. They were then decalcified in five days with TBD-2™ Decalcifier, Epredia™ and processed with the Tissue Processor™ (Leica Microsystems, Mannheim, Germany) and Thermo Scientific™ Histoplast Paraffin™ equipment. Finally, for histology studies, the samples were cut to 5 µm with the Leica microtome equipment (Leica Microsystems Mannheim, Germany). The sections were stained with Masson’s Trichrome (MT) and Hematoxylin-Eosin (H&E) stains to be analyzed by a Leica microscope with the Leica suite application for imaging (Leica Microsystems, Mannheim, Germany). 

The surface morphology of the samples and their respective elemental analysis was assessed using a scanning electron microscope (SEM) JEOL Model JSM 6490 LV coupled to an X-ray Energy Dispersion Spectrometer (EDS) (Akishima, Tokyo, Japan) after metalized coating on its surface with gold using the Denton Vacum Model Desk IV equipment. Finally, the guidelines for Animal Research: Reporting in vivo experiments (ARRIVE) were considered [22]. The protocols applied in this research were based on the ethical principles for animal research adopted by the LABBIO Laboratory of the Universidad del Valle (Cali-, Colombia). The Ethics Committee approved them with Biomedical Experimental Animals (CEAS) from the Universidad del Valle (CEAS 012-019).

## 3. Results

### 3.1. Characterization of the Synthesized Chitosan Beads

We previously discussed the preparation and characterization of the CS beads (Table 1) by Fourier Transformed Infrared spectroscopy (FT-IR), X-ray diffraction (XRD), scanning electron microscopy (SEM), and thermogravimetric analysis (TGA) [18]. The characterization of the ethanolic extract used to prepare the F5 presented the main component, cyanidin-3-rutinoside, as shown by ultra-high-performance liquid chromatography (UHPLC, Appendix A) [21]. Briefly, the FT-IR spectrum confirmed the crosslinking reaction of CS with GLA through the band at 1641 cm^−1,^ usually attributed to the imine-like bond from the amino groups of CS and the carbonyl groups of GLA. On the other hand, the XRD analysis for the formulations including TiO_2_ showed characteristic angles 2θ = 25°, 38°, 48°, 54°, 62°, 68°, 74°, and 82°, of the anatase phase of TiO_2_. The CS/GLA/GO/TiO_2_ beads presented a small peak at 2θ = 13° attributed to GO intercalation. SEM microstructure studies demonstrated alterations in the surfaces of the CS/GLA beads, increasing the roughness when GO and TiO_2_ were added due to the intercalation of the nanomaterials with the polymeric chains.

### 3.2. Critical Bone Defect Studies

In orthopedic, trauma, and maxillofacial surgery procedures, bone tissue grafts (autografts) or bone substitutes in patients to regenerate or rebuild lost tissue [23]. Most of the bone substitutes currently available commercially come from mainly dead human donors (allograft) or from animals (xenograft), which tend to be rejected by patients because of their origin or because of the belief that they can transmit diseases [24]. In our research, the possibility of manufacturing bone substitutes of a synthetic origin (polyvinyl alcohol) combined with a safe and biocompatible polysaccharide (chitosan), which was also reinforced with graphene oxide and titanium dioxide nanoparticles, was explored to improve durability without sacrificing the biocompatibility of the beads. Blackberry residue extract was introduced to provide anthocyanins that improve biocompatibility and provide antioxidant characteristics. This system has not been reported so far for applications as a bone substitute in critical bone defects. This approach, if successful, will make it possible to end the dependence on grafts of human origin, which depend on the donation made by patients or family members before they die. Likewise, it could be a good alternative to grafts of animal origin. 

Under the ISO 10993-6 standard, a macroscopic inspection of the biomodels and samples was carried out in search of alterations related to inflammation or necrosis. Figure 1 corresponds to the recovery of the samples after 90 days of implantation. The figure shows how the hair in the cranial area was completely recovered. After the trichotomy, the skin presented a healthy appearance without scars or inflamed areas. Besides, after tissue separation accessing the intervened area, it was easy to appreciate that all the beads were covered with soft transparent tissues in the bone area without necrosis. 

The macroscopic observation of hair recovery and the absence of inflammation or necrotic tissues is the first indication of biocompatibility, because the implanted materials allowed the tissues to heal generally without interfering in this process. The macroscopic observation of the tissues is a recommendation of the iso 10993-6 standard, and the finding of good skin conditions after implantation of compatible materials has been reported by other authors [25].

On the other hand, the microscopic images of the samples recovered with bone fragments are observed in Figure 2. In all cases, the samples were covered by a soft scar tissue of a transparent appearance that seems to integrate with the periosteum. The periosteum is a part of the soft tissue (skin, gum, or oral mucosa) that covers the bone tissue and can repair it when injured. Our investigation’s histological results showed that the intraosseous preparation healing is carried out at the expense of a fibrous tissue composed of bundles of type I collagen fibers that seem to continue with the periosteum, which seems to be to indicate typical intraosseous scarring. What is interesting is that type I collagen is one of the main components of the extracellular bone matrix, and the process is occurring in a critical size defect, which would not be expected to regenerate unless the implanted material has the property of activating this process.

However, little reabsorption of the implanted materials is observed without the presence of a severe or necrotic immune response. The minimum reabsorption observed in the implanted beads is due to the material’s stability, and this guarantees its presence during the creeping substitution process, which ranges between six and twelve months to have a partial replacement of the material [26], and years for a complete replacement, depending on the type of material used [27]. In the case of the critical size design in parietal bone, it would be expected that the reabsorption of the implanted materials will be difficult due to the characteristic of the parietal bone, which is very cortical and therefore not very vascularized [28].

In this work, the critical size defect experimental design was applied. The critical size of the defect has been defined as an intraosseous preparation that did not heal spontaneously during the experiment [29] and is considered an ideal scenario to evaluate biomaterials that may have utility in bone regeneration [30]. In Figure 3, it is possible to observe how in the control samples (empty defect), the preparation is also covered by a soft tissue with a transparent appearance, without evidence of regeneration of the bone defect. The absence of regeneration of the control defect agrees with what was expected because this type of preparation should not show an appreciable regeneration during the duration of the experiment, which in this case was 90 days [31].

Using the SEM technique (Figure 4), it was identified that the covering soft tissue is made up of fibrous bundles, and in some places, the integration between the tissue that covers the beads and the periosteum that covers the surrounding bone tissue, is appreciated. Other authors have already reported the bundles of collagen fibers that cover the surgical preparations, but especially in histological studies with trichrome staining [32], which are also reported as a fibrous membrane covering the preparations [33].

Histological and histochemical studies also confirmed soft tissue covering the implanted areas and individually surrounding each bead. The first column of images in Figure 5 shows the five types of beads surrounded by soft tissue that appears to integrate with the periosteum of the adjacent bone. Utilizing Masson’s Trichromacy staining confirmed that this tissue corresponds to collagen type I and surrounds each bead individually. However, it joins the bone tissue, having continuity with the periosteum, which has also been reported in implanting other compatible materials [34].

In a critical size defect, it would not be expected to find bone tissue formation unless the material that is implanted has an osteogenic potential as has been reported in other studies in which materials such as calcium phosphates have been implanted [35], or as we reported previously with graphene oxide [36]. In this investigation, the presence of newly formed bone tissue was found in the interface area between the grains and the surrounding bone tissue, corresponding to the area of F5 implantation (Figure 5Ñ), where it can be seen that these structures have a slightly different appearance compared with the tissue surrounding the bone, indicating a lower level of maturity (Figure 6). 

Figure 6 corresponds to the image of F5, where newly formed bone tissue was evidenced. The image clearly shows that the newly formed bone tissue is not fully mature and occupies the interface area, filling the space between the bead and the muscle and continuing with the peripheral bone tissue. This finding contrasts with the results obtained with the other formulations with which bone tissue formation was not evidenced in the areas of the preparations. The presence of bone tissue in the implantation area of this specific material would indicate that the implanted material has stimulated the regeneration of the critical size defect, since it is accepted that this type of preparation does not regenerate spontaneously [37]. 

Masson’s trichromacy technique is accepted in histology as specific staining to identify collagen fibers in connective tissues [38]. Figure 5, for example, shows solid blue staining that is explained because the organic component of the bone extracellular matrix mainly is collagen Type I [39].

The presence of collagen fibers in the bone defect, surrounding the beads and occupying the space between them, is a finding that could be related to mineralization if the relationship between fibrillogenesis and mineralization is taken into account, because the collagen fibers present in the extracellular bone matrix are the scaffold used by osteoblasts as a support for mineralization [40]. 

Size defects are an experimental design widely used to test materials that show potential bone regeneration applications. This design is characterized by the preparation of intraosseous defects that, due to their dimensions, do not allow spontaneous regeneration through new bone formation [31]. In this way, everything observable in the experiment can be attributed to the properties of the material. In this work, the empty defect did not show evidence of regeneration, remaining as a preparation covered by tissue similar to the periosteum. 

The periosteum is the tissue that lines the bone surfaces and is considered to have great osteogenic potential. Here, the periosteum is removed to perform intraosseous preparations; however, it recovers and recoats the surfaces when healing occurs. This can be observed when reviewing histological images of similar works, in which the periosteum covers both the neighboring bone tissue and the intervening areas [41].

Despite the tremendous osteogenic capacity of the periosteum and its role in the regeneration of other types of bone defects [42], the potential of this tissue is not sufficient to stimulate the regeneration of the control preparations (empty defects), which are the main characteristic of this experimental design. Usually, when a material with osteogenic potential is implanted, periosteum cells can contribute to the healing of the defect [43]. 

In this work, the control preparations remained empty, while, of the experimental defects, only F5 showed new bone formation at the preparation site (Figure 6). The formation of new bone seems to be stimulated by the additional presence of the anthocyanin cyanidin-3-rutenoside that could function as an elicitor to produce growth factors or as a gene activator for cell differentiation. Anthocyanins have previously been shown to benefit human health (anticancer activity, anti-inflammatory activity, neuroprotective activity, prevention of cardiovascular disease, anti-obesity, and anti-diabetic activities) [44]. Anthocyanins stimulate nuclear reprogramming through the increased transcription factor expression, which could also stimulate cell differentiation [45]. For this reason, this study is of great interest, adding value to waste from a pulping company that does not re-use this waste.

SEM-EDS analysis was performed for all the formulations used in the present study. Figure 7 shows the elemental analysis, confirming the presence of calcium in F3 and F5. However, F5 also has phosphorus, which is an essential component in the mineralization process. As previously stated, the mineralization process might be stimulated for the anthocyanin present in the blackberry waste extract. 

## 4. Conclusions and Future Perspectives

This research found that the materials behaved as compatible in the five formulations, allowing soft tissue healing with hair recovery and the absence of necrotic areas. At the level of intraosseous implantations, the materials were covered by soft tissue with a transparent appearance that seems to come from the periosteum and histologically is made up of bundles of collagen type I.

Histologically, the presence of newly formed bone tissue could only be identified in the sample corresponding to F5. The presence of calcium (~10%) in F3 can be explained by the beginning of mineralization and extracellular bone matrix formation. Moreover, calcium and phosphorus (~1%) in F5 indicates a maturation process of the bone matrix, stimulated by the anthocyanin from the BBE.

The results obtained with the formulations F3 and F5 concerning the new extracellular material deposited observed with SEM indicate potential use as a bone substitute. However, it is imperative to assess a study with a more significant number of biomodels to quantify the newly bone matrix formed area and the percentage of calcium deposited.

The results of this research are auspicious for the development of bone substitutes and evidence that the technologies for tissue regeneration, including chitosan nanocomposites, are beneficial for the adhesion and proliferation of bone cells.

## Figures and Tables

**Figure 1 polymers-13-03877-f001:**
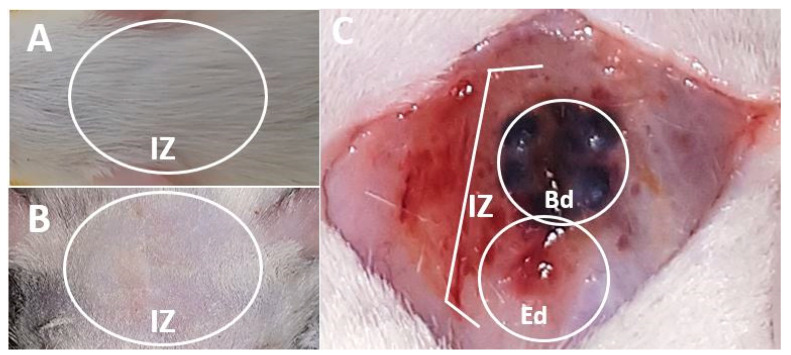
Macroscopic image of the skull intervened area. (**A**): Presence of hair. (**B**): Trichotomy. (**C**): Exposed bone area. IZ: Implantation zone. Bd: Beads. Ed: Empty defect.

**Figure 2 polymers-13-03877-f002:**
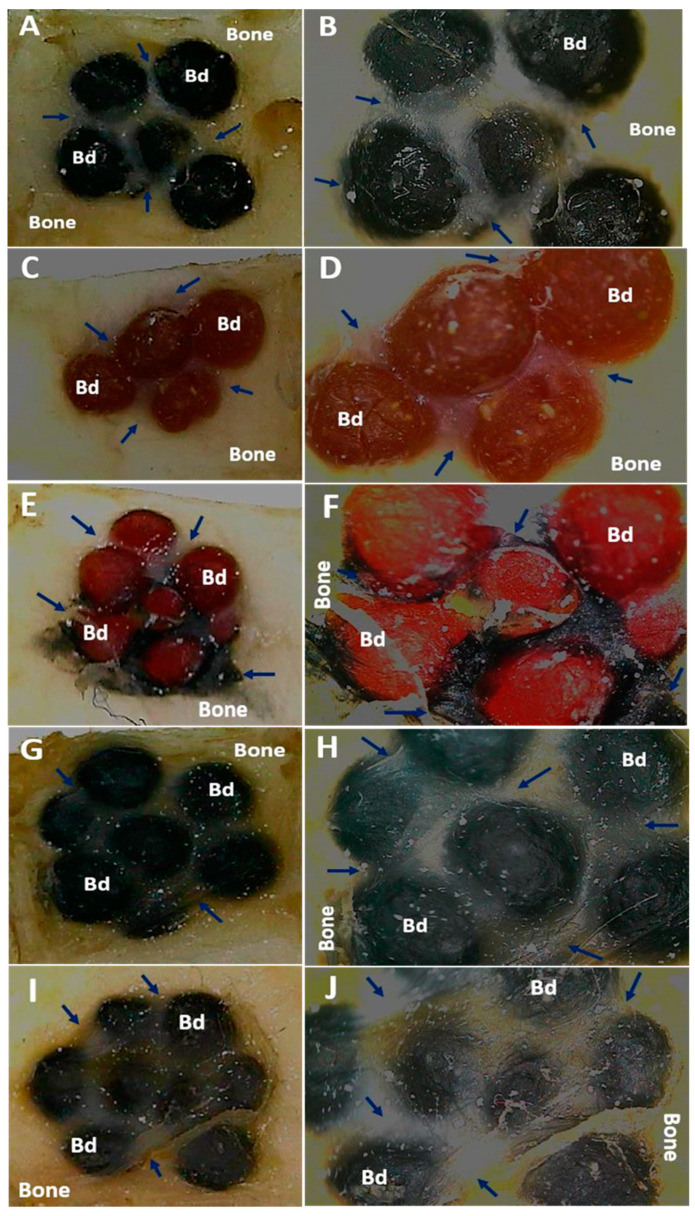
Cranial implantation areas. (**A**): F1 at 2×. (**B**): F1 at 3×. (**C**): F2 at 1×. (**D**): F2 at 3×. (**E**): F3 at 1×. (**F**): F3 at 3×. (**G**): F4 at 1×. (**H**): F4 at 3×. (**I**): F5 at 1×. (**J**): F5 at 3×. Bd: Bead. Blue arrows: soft tissue surrounding intervened areas. Stereoscopic microscope technique.

**Figure 3 polymers-13-03877-f003:**
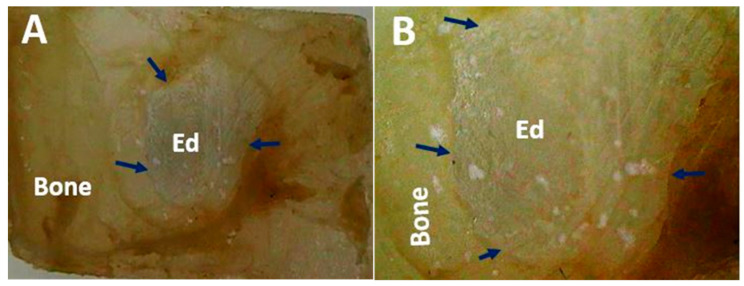
Cranial implantation area of the empty defect. (**A**): Preparation at 1×. (**B**): Preparation at 3×. Blue arrows: soft tissue covering. Stereoscopic microscope technique.

**Figure 4 polymers-13-03877-f004:**
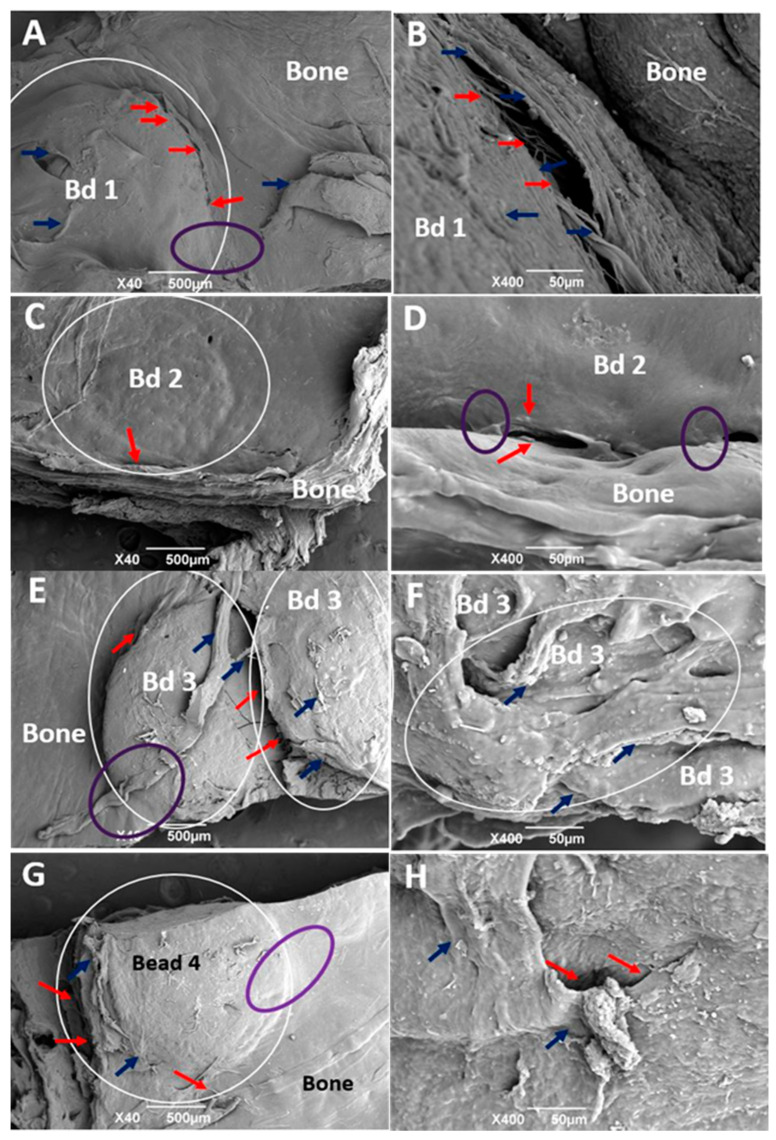
SEM analysis of the cranial implantation areas. (**A**,**B**): F1. (**C**,**D**): F2. (**E**,**F**): F3. (**G**,**H**): F4.. White circle: Bead implantation area. Red arrow: Interface area bead–bead, bead–bone. Blue arrow: Cover soft tissue. Purple oval: Incorporation area of the bead covering soft tissue with the bone covering soft tissue.

**Figure 5 polymers-13-03877-f005:**
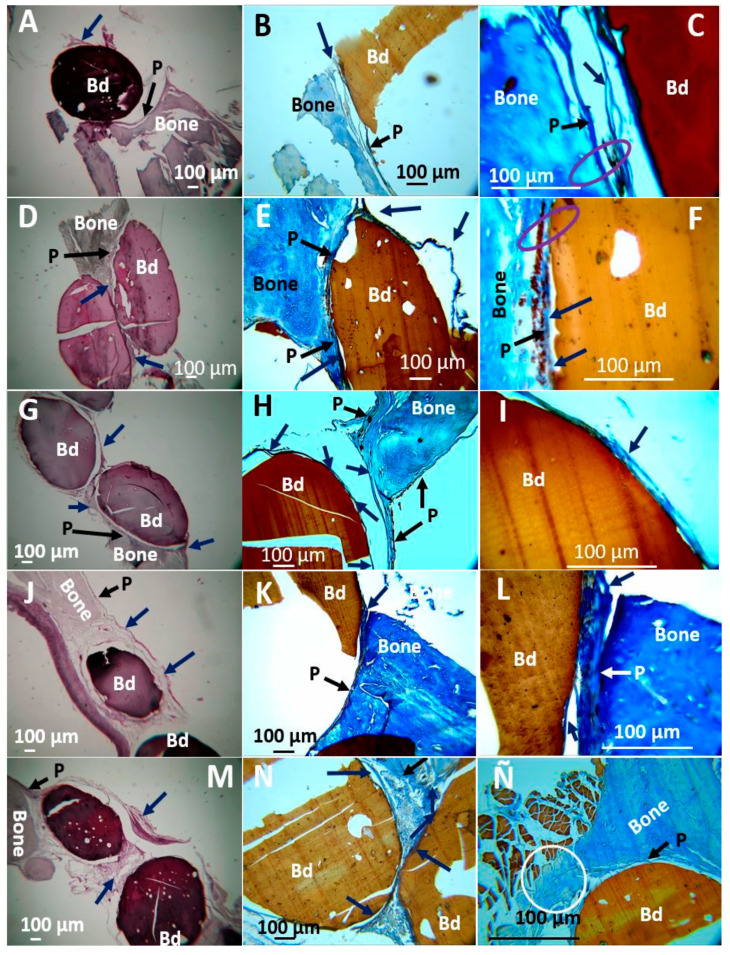
Samples implanted in the cranial bone defect. (**A**–**C**): F1. (**D**–**F**): F2. (**G**–**I**): F3. (**J**–**L**): F4. (**M**–**Ñ**): F5. (**A**,**D**,**G**,**J**,**K**) at 4 × H-E technique. (**B**,**E**,**H**,**K**,**N**) at 10 × MT technique. (**C**,**F**,**I**,**L**,**Ñ**) at 40 × MT technique. Bd: Bead. Black arrow P: Periosteum. Blue arrow: Soft tissue that covers the beads. Purple Oval: Areas where there is continuity of bone-bead soft tissue. White Circle: area of new bone formation.

**Figure 6 polymers-13-03877-f006:**
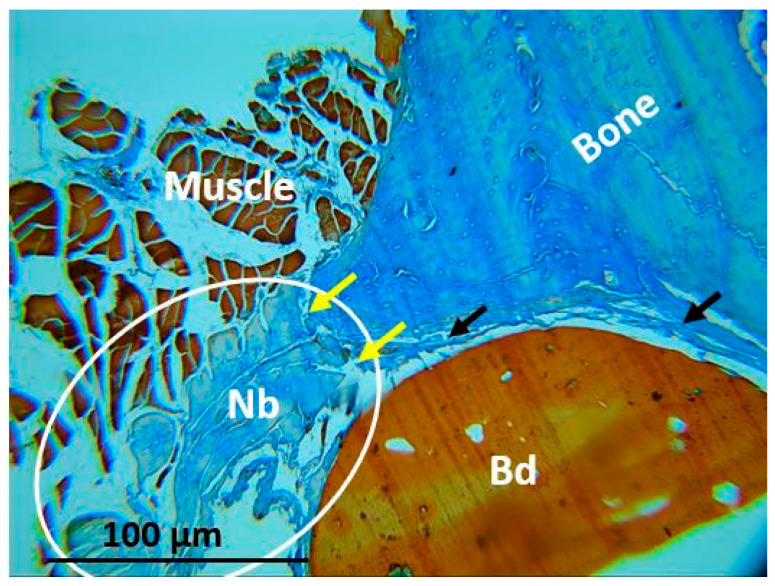
Implantation area for F5. White circle: Area of newly formed bone. NB: Newly formed bone. Yellow arrow: Interface between the newly formed bone and the surrounding bone. Bd: Bead.

**Figure 7 polymers-13-03877-f007:**
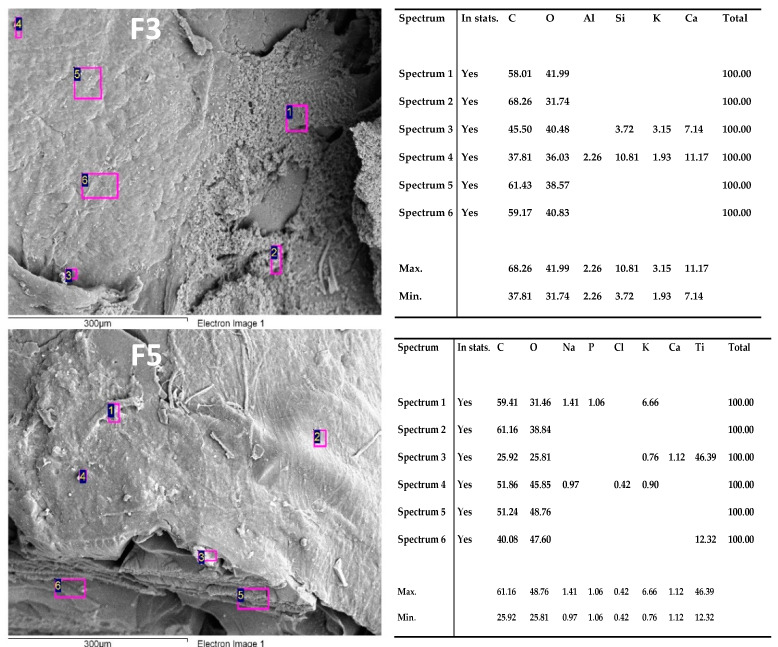
SEM-EDS analysis for F3 and F5 formulations inside after bone implantation studies.

**Table 1 polymers-13-03877-t001:** Composition of chitosan beads for the critical bone defect study.

Name	System	Formulation
F1	CS/GLA	Chitosan, glutaraldehyde
F2	CS/GLA/GO	Chitosan, glutaraldehyde, graphene oxide
F3	CS/GLA/TiO_2_	Chitosan, glutaraldehyde, titanium dioxide nanoparticles
F4	CS/GLA/GO/TiO_2_	Chitosan, glutaraldehyde, graphene oxide, titanium dioxide nanoparticles
F5	CS/GLA/GO/TiO_2_/BBE	Chitosan, glutaraldehyde, graphene oxide, titanium dioxide nanoparticles, blackberry waste extract

## Data Availability

Data available under request to the corresponding author.

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
