# Peer review of "Nanocomposites of Chitosan/Graphene Oxide/Titanium Dioxide Nanoparticles/Blackberry Waste Extract as Potential Bone Substitutes"

_polymers, 2021, doi:10.3390/polym13223877_

Round 1

Reviewer 1 Report

Greetings, Editor thank you for providing me with the opportunity to review the article. I reviewed the article with title ``  Nanocomposites of Chitosan/Graphene Oxide/Titanium Dioxide Nanoparticles/BlackBerry waste extract as Potential Bone  Substitutes ``.  The article topic is intriguing and promising in the area. Overall, the article content and quality are suitable for the polymer journal. I am pleased to send you major level comments. Please consider these suggestions as listed below.

  1. The abstract is written well but please concise the abstract, there are detailed information about results. Please present an outlook of results, rest detail is already in manuscript.
  2. Introduction section must be written on more quality way, i.e. more up-to-date references addressed.
  3. Please do not use lumpy reference such as [2-6, 16-23, 34-38] etc. Each reference needs to be properly addressed. Please revise your paper accordingly since same issue occurs on several spots in the paper. Please delete reference from 16-23 and cite only this article -Recent advances in metal decorated nanomaterials and their various biological applications: a review.
  4. The novelty of the work must be clearly addressed and discussed, compare your research with existing research findings and highlight novelty, (compare your work with existing research findings and highlight novelty).
  5. Research gap should be delivered on more clear way with directed necessity for the conducted research work.
  6. The main objective of the work must be written on the more clear and more concise way at the end of introduction section. It seems weird in present form.
  7. Please remove the 31-34 reference number and cite this only one ``Yaqoob, A. A., Ahmad, A., Ibrahim, M. N. M., & Rashid, M. (2021). Chitosan-based nanocomposites for gene delivery: Application and future perspectives. In Polysaccharide-Based Nanocomposites for Gene Delivery and Tissue Engineering (pp. 245-262). Woodhead``.
  8. What is CS in Materials section?
  9. What is periosteum?
  10. Please provide high resolution quality of Figure 7.
  11. The Results and Discussion section are quite well written and presented; I really appreciated the efforts. However, regarding the replications, authors confirmed that replications of experiment were carried out. However, these results are not shown in the manuscript, how many replicated were carried out by experiment? Results seem to be related to a unique experiment. Please, clarify whether the results of this document are from a single experiment or from an average resulting from replications. If replicated were carried out, the use of average data is required as well as the standard deviation in the results and figures shown throughout the manuscript. In case of showing only one replicate explain why only one is shown and include the standard deviations.
  12. Please check the abbreviations of words throughout the article. All should be consistent.
  13. Section 4 should be renamed by Conclusion and Future perspectives. Conclusion section is missing some perspective related to the future research work, quantify main research findings, highlight relevance of the work with respect to the field aspect.
  14. Please strictly follow the journal guidelines, there are several formatting errors.
  15. To avoid grammar and linguistic mistakes, minor level English language should be thoroughly checked. 

Author Response

We thank the reviewer for all the valuable suggestions and corrections that help improve the manuscript's quality. All the authors very much appreciate your time and effort in doing so. All the indications are highlighted in the attached document and also in the manuscript for your convenience

Reviewer 2 Report

The article is interesting and well written. I would like to know, if the studied chitosan-based materials were also tested in vitro. In my opinion this kind of study would help to explain their influence on the tissue forming. Also, I know that the studied materials were already characterized and the results are presented in another paper, but I think that some information about their composition (eg. the content of the ingredients) and properties should also be given in this paper, to make it more clear.

Author Response

(The authors gave the same response as above.)

Round 2

Reviewer 1 Report

Dear Authors
I have reviewed again the manuscript and I think that it is ready for publication. Thank you for considering my suggestions